# Consistent condom use among men who pay for sex in sub-Saharan Africa: Empirical evidence from Demographic and Health Surveys

Bright Opoku Ahinkorah[1], Eugene Budu[2], Abdul-Aziz Seidu[2,3], John Elvis Hagan, Jr.[4,5]*, Ebenezer Agbaglo[6], Thomas Hormenu[4], Thomas Schack[5], Sanni Yaya[7,8]

1 The Australian Centre for Public and Population Health Research (ACPPHR), Faculty of Health, University of Technology Sydney, Sydney, Australia, 2 Department of Population and Health, University of Cape Coast, Cape Coast, Ghana, 3 College of Public Health, Medical and Veterinary Sciences, James Cook University, Townsville, Queensland, Australia, 4 Department of Health, Physical Education, and Recreation, University of Cape Coast, Cape Coast, Ghana, 5 Neurocognition and Action-Biomechanics-Research Group, Faculty of Psychology and Sport Sciences, Bielefeld University, Bielefeld, Germany, 6 Department of English, University of Cape Coast, Cape Coast, Ghana, 7 School of International Development and Global Studies, University of Ottawa, Ottawa, Canada, 8 The George Institute for Global Health, The University of Oxford, Oxford, United Kingdom

* elvis.hagan@ucc.edu.gh

## Abstract

### Background

Paying for sex has often been associated with risky sexual behavior among heterosexual men, and men who pay for sex are considered as a bridging population for sexually transmitted infections. Consistent condom use during paid sex is essential for reducing sexually transmitted infections, including HIV/AIDS. In this study, we assessed the prevalence and predictors of consistent condom use among men who pay for sex in sub-Saharan Africa.

### Materials and methods

We pooled data from 29 sub-Saharan African countries' Demographic and Health Surveys. A total of 3,353 men in sub-Saharan Africa who had paid for sex in the last 12 months preceding the surveys and had complete information on all the variables of interest were used in this study. The outcome variable for the study was consistent condom use for every paid sex in the last 12 months. Both bivariate and multivariable logistic regression analyses were carried out. Results were presented as adjusted odds ratios with their corresponding 95% confidence intervals. Statistical significance was declared at p< 0.05.

### Results

Overall, the prevalence of consistent condom use during paid sex in sub-Saharan Africa was 83.96% (CI = 80.35–87.56), ranging from 48.70% in Benin to 98% in Burkina Faso. Men aged 35–44 [AOR, 1.39 CI = 1.04–1.49], men in the richest wealth quintile [AOR, 1.96 CI = 1.30–3.00], men with secondary level of education [AOR, 1.69 CI = 1.17–2.44], and

**Data Availability Statement:** All relevant data are within the manuscript. The dataset is freely

accessible at: https://dhsprogram.com/data/available-datasets.cfm.

**Funding:** We sincerely thank the German Research Foundation through the Neurocognition and Action-Biomechanics Research Group, Bielefeld University, Germany for providing financial support for the publication of this research.

**Competing interests:** The authors have declared that no competing interests exist.

men in Burkina Faso [AOR = 67.59, CI = 8.72–523.9] had higher odds of consistent condom use during paid sex, compared to men aged 15–19, those in the poorest wealth quintile, those with no formal education, and men in Benin respectively. Conversely, Muslim men had lower odds [AOR = 0.71, CI = 0.53–0.95] of using condom consistently during paid sex, compared to Christian men.

## Conclusion

Empirical evidence from this study suggests that consistent condom use during paid sex encompasses complex social and demographic characteristics. The study also revealed that demographic characteristics such as age, wealth quintile, education, and religion were independently related to consistent condom use for paid sex among men. With sub-Saharan Africa having the highest sexual and reproductive health burden in the world, continuous application of evidence-based interventions (e.g., educational and entrepreneurial training) that account for behavioural and social vulnerabilities are required.

## Introduction

Consistent condom use during paid sex is essential for reducing sexually transmitted infections (STIs), including HIV/AIDS [1,2]. HIV/AIDS has been one of the greatest contributors to the global mortality rate [3,4]. As indicated by UNAIDS [4], globally, 37.9 million [32.7–44.0 million] people were living with HIV at the end of 2018. An estimated 0.8% [0.6–0.9%] of adults aged 15–49 years worldwide are living with HIV, although the burden of the epidemic continues to vary considerably between countries and regions. Sub-Saharan Africa (SSA) remains most severely affected, with 1 in every 25 adults (3.9%) living with HIV and accounting for more than two-thirds of the people living with HIV worldwide. In 2018, that number reached 25.7 million [22.2–29.5], accounting for nearly 71% of the world's total HIV-infected individuals, and around 75% of HIV-induced deaths in the sub-region [5]. More than half of the world's new HIV infections also occur in SSA, particularly in Eastern and Southern Africa, which record nearly 42.5% of all new cases worldwide [6,7]. This figure suggests that the burden of HIV epidemic disproportionately affects SSA. Within SSA, and the world at large, HIV transmission is often facilitated by men who pay for sex, as such men serve as a 'bridge' for HIV transmission through having unprotected sex with their female clients, spouses, girlfriends, men, and others [8]. Paying for sex involves the exchange of sex for money, gift, services, or other favours such as promotion at the work place and grades in school [9–12].

The global community has proposed for the end of the HIV pandemic, as evident in the then Millennium Development Goal 6 (i.e., combating HIV/AIDS, malaria, and other diseases) and the current Sustainable Development Goal 3 (i.e., ensuring healthy lives and promoting wellbeing for all at all ages) [13]. UNAIDS has championed and aimed at remarkable reduction of both HIV infections and deaths by the year 2030. This notwithstanding, the literature suggests that, in recent times, there has been a decline in global spending on HIV in SSA [14]. This trend calls for a revival of commitments at reducing the HIV pandemic, especially in the sub-region [15] through the adoption of preventive measures. One of such preventive strategies is consistent use of condoms [16]. Consistent condom use, as a means of reducing HIV infections, has been identified as a public health priority and a critical component in prevention programs [17,18]. Aside the use of preventive measures, there is the need to extend

research on HIV/AIDS [15], as such research is needed for public health interventions aimed at ending the pandemic.

Globally, research on condom use among men has mainly been conducted in Asian countries such as Cambodia [19], China [20,21], India [22–24], Singapore [25], Indonesia [26], with a few focusing on Middle East [1] and Africa [27,28]. Generally, such studies have revealed the predictors of consistent condom use among homosexual males [19,27,29], young dating adults [30], and men who patronize paid sex [22]. For SSA, such studies have often been country-level analysis, with the focus on Togo [27], Nigeria [28], and Ghana [31]. With the adoption of country-specific focus, such studies could not provide a panoramic view of the reality of consistent condom use in the sub-region. Additionally, predictors of consistent condom use among men who patronize paid sex are yet to be explored in the sub-region. To fill these gaps, the present study assessed the prevalence and predictors of consistent condom use among men who pay for sex in SSA. Findings from this study add to the existing literature to help in fighting against HIV/AIDs and other STIs in SSA through empirical data, and might provide a baseline information for designing appropriate interventions.

## Methods and materials

### Data source

We pooled data from 29 SSA countries' Demographic and Health Surveys (DHS) with information on paid sex. Specifically, we used data from the men's file from the various countries. The DHS is a nationally representative survey that is conducted in over 85 low- and middle-income countries globally. The survey focuses on essential maternal and child health markers and men's health, including paid sex and consistent condom use [32]. The survey employs a two-stage stratified sampling technique, which makes the survey data nationally representative [33]. The dataset is freely accessible at https://dhsprogram.com/data/available-datasets.cfm. Details of the DHS methodology have been reported in previous studies [11,32]. A total of 3,353 men in SSA who had paid for sex in the last 12 months preceding the survey and had complete information on all the variables of interest were used. We followed the 'Strengthening the Reporting of Observational Studies in Epidemiology' (STROBE) statement in the conduct of the current study.

### Definition of variables

**Outcome variable.**   The outcome variable for the study was consistent condom use for every paid sex in the last 12 months. It was derived from the question, "Did you use a condom every time you paid for sex in the last 12 months?". The responses were "Yes" and "No," which were coded 1 and 0 respectively.

**Explanatory variables.**   The study made use of nine explanatory variables that had shown significant associations with consistent condom use during paid sex in previous studies [22,34–36]. These variables were place of residence, wealth quintile, education, age, frequency of reading newspaper, frequency of listening to radio, frequency of watching television, religion, marital status, and employment (see Table 1).

**Statistical analyses.**   The data was analysed with STATA version 14.0. The analysis was done in three steps. First, the computation of consistent condom use among men who pay for sex in SSA was done. This procedure was done by generating a forest plot, using the syntax "metaprop" in STATA version 14.0 (StataCorp, College Station, TX, USA). The forest plot showed the prevalence of consistent condom use during paid sex in individual countries and the pooled prevalence of consistent condom use during paid sex in all the countries with their associated 95% confidence intervals and corresponding weight. Before this procedure, a test of

**Table 1. Socio-demographic characteristics and consistent condom use during paid sex among men in SSA (Weighted).**

| Variable | Frequency (n) | Percentage (%) | Consistent condom use during paid sex | χ2 (p-value) |
|---|---|---|---|---|
| **Residence** | | | | 27.2 (<0.001) |
| Urban | 1644 | 49.0 | 51.1 | |
| Rural | 1709 | 51.0 | 48.9 | |
| **Wealth quintile** | | | | 58.0 (<0.001) |
| Poorest | 457 | 13.6 | 12.3 | |
| Poorer | 546 | 16.3 | 15.2 | |
| Middle | 638 | 19.0 | 18.9 | |
| Richer | 775 | 23.1 | 23.6 | |
| Richest | 937 | 28.0 | 30.0 | |
| **Education** | | | | 46.4 (<0.001) |
| No education | 331 | 9.9 | 8.6 | |
| Primary | 1086 | 32.4 | 31.5 | |
| Secondary | 1675 | 50.0 | 51.7 | |
| Higher | 261 | 7.8 | 8.3 | |
| **Age** | | | | 8.73(0.039) |
| 15–24 | 1202 | 35.8 | 35.0 | |
| 25–34 | 1282 | 38.2 | 39.0 | |
| 35–44 | 626 | 18.7 | 19.0 | |
| 45+ | 243 | 7.3 | 7.0 | |
| **Frequency of reading newspaper/magazine** | | | | 16.1(<0.001) |
| Not at all | 2019 | 60.2 | 58.7 | |
| Less than once a week | 678 | 20.2 | 21.1 | |
| At least once a week | 656 | 19.6 | 20.2 | |
| **Frequency of listening to radio** | | | | 17.3(<0.001) |
| Not at all | 841 | 25.1 | 23.7 | |
| Less than once a week | 697 | 20.8 | 20.8 | |
| At least once a week | 1815 | 54.1 | 55.5 | |
| **Frequency of watching television** | | | | 30.4(<0.001) |
| Not at all | 1322 | 39.4 | 37.4 | |
| Less than once a week | 677 | 20.2 | 20.4 | |
| At least once a week | 1354 | 40.4 | 42.2 | |
| **Religion** | | | | 12.0(0.007) |
| Christianity | 2105 | 62.8 | 63.8 | |
| Islam | 893 | 26.6 | 25.5 | |
| Other | 92 | 2.7 | 2.7 | |
| No religion | 263 | 7.9 | 8.0 | |
| **Marital status** | | | | 1.2(0.882) |
| Never married | 1598 | 47.6 | 47.5 | |
| Married | 1012 | 30.2 | 30.1 | |
| Cohabiting | 428 | 12.8 | 12.9 | |
| Widowed | 27 | 0.8 | 0.9 | |
| Divorced | 288 | 8.6 | 8.7 | |
| **Employment** | | | | 33.4(<0.001) |
| Not working | 336 | 10.0 | 10.1 | |
| Managerial | 273 | 8.1 | 8.6 | |
| Clerical | 61 | 1.8 | 1.9 | |
| Sales | 390 | 11.6 | 11.7 | |

*(Continued)*

**Table 1.** (Continued)

| Variable | Frequency (n) | Percentage (%) | Consistent condom use during paid sex | χ2 (p-value) |
|----------|---------------|----------------|----------------------------------------|--------------|
| Agriculture | 866 | 25.8 | 23.9 | |
| Services | 344 | 10.3 | 10.4 | |
| Manual | 1083 | 32.3 | 33.3 | |

heterogeneity of the DHS data obtained for the countries was done. A high level of inconsistency ($I^2 > 50\%$) was identified. This obtained value supported the use of a random effect model in meta-analysis. The second step was a bivariate analysis that calculated the prevalence of consistent condom use for every paid sex in the last 12 months across the socio-demographic characteristics of men with their significance levels. Additional test for multicollinearity was conducted and no evidence of collinearity among the independent variables (Mean VIF = 1.32, Maximum VIF = 1.83, Minimum = 1.01) was found. All the variables that showed statistical significance at the bivariate level were moved to the multivariable level. The third step was the multivariable analysis using binary logistic regression modelling. Results were presented as adjusted odds ratios with their corresponding 95% confidence intervals to indicate their level of precision. The choice of reference categories at the regression analysis stage was informed by a priori, variable outcome categories with lower frequencies, and previous studies [22,34–36]. Statistical significance was set at $p < 0.05$. Sample weight was applied and the survey command (svy) was used to account for the complex sampling design of the survey.

**Ethical approval.** Ethical clearance was obtained from the Ethics Committee of ORC Macro Inc. as well as Ethics Boards of partner organisations of the various countries such as the Ministries of Health. The DHS follows the standards for ensuring the protection of respondents' privacy. Inner City Fund International ensures that the survey complies with the U.S. Department of Health and Human Services regulations for the respect of human subjects. The survey also reported that both verbal and written informed consent were obtained from the respondents. However, the secondary analysis of data employed meant that no further approval was required for this study. Further information about the DHS data usage and ethical standards are available at http://goo.gl/ny8T6X.

## Results

### Proportion of men who consistently use condom during paid sex in SSA

Fig 1 shows results on the proportion of men who consistently use condom during paid sex in SSA. Overall, prevalence of consistent condom use during paid sex in SSA was 83.96% (CI = 80.35–87.56), ranging from 48.70% (39.64–57.76) in Benin to 98% (94.12–101.88) in Burkina Faso.

### Bivariate analysis on the socio-demographic characteristics and consistent condom use during paid sex among men in SSA

Table 1 shows results on the distribution of consistent condom use during paid sex across the socio-demographic characteristics of the respondents. The results indicate that consistent condom use during paid sex was high among men who lived in urban areas (51.1%), richest men (30%), those with secondary level of education (51.7%), those aged 25–34 years (39%), those who did not read newspaper/magazine (58.7%), those who listened to radio at least once a week (55.5%), those who watched television at least once a week (42.2%), Christians (63.8%), those who were never married (47.5%), and those in manual employment (33.3%). The results

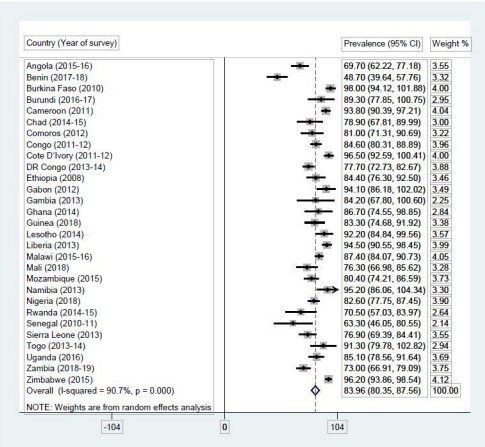

**Fig 1. Proportion of men who consistently use condom during paid sex in SSA.**

further show that all the explanatory variables except marital status had statistically significant associations with consistent condom use during paid sex.

## Multivariable logistic regression analysis on the factors that influence consistent condom use among men who pay for sex in SSA

Table 2 shows results of the binary logistic regression analysis on the factors that influence consistent condom use among men who pay for sex in SSA. The results reveal that consistent condom use among men who pay for sex increases with age, with men aged 35–44 more likely to use condom consistently during paid sex than those aged 15–24 [AOR, 1.39 CI = 1.04–1.49]. Across wealth quintile, highest odds of consistent condom use among men who pay for sex was found among men with richest wealth quintile [AOR, 1.96 CI = 1.30–3.00]. Men with secondary level of education were more likely to use condom consistently during paid sex, compared to those with no level of education [AOR, 1.69 CI = 1.17–2.44] while Muslim men were less likely to use condom consistently during paid sex, compared to men who were Christians [AOR = 0.71, CI = 0.53–0.95]. From the surveyed countries, men who lived in Burkina Faso had the highest odds of consistent condom use during paid sex, compared to men who lived in Benin [AOR = 67.59, CI = 8.72–523.9].

## Discussion

The current study adds to the sexuality education literature on men by assessing the correlates of consistent condom use among men who pay for sex in SSA. Findings would help expound on how interpersonal characteristics of men could impact condom use during paid sex. Across the studied countries, overall, consistent condom use among men who paid for sex in SSA was high (83.96%), ranging from 48.70% in Benin to 98% in Burkina Faso (see Fig 1). With the varying proportions of consistent condom use across the studied countries, this finding emphasizes the need for continuous attention on increasing consistent condom use among people engaging in risky sexual behaviours, especially men with females in paid sex work, because of the high risk associated with their casual and/ or multiple partnerships.

Because of the fear of high risk of HIV infection and other STIs, paid sex workers may consistently demand or negotiate condom usage with their male clients [37,38]. Additional empirical evidence reveals that, in countries where paid sex workers have limited means of other

**Table 2. Multivariable logistic regression analysis on the factors that influence consistent condom use among men who pay for sex in SSA.**

| Variables | AOR | 95% CI | |
|---|---|---|---|
| | | Lower Bound | Higher Bound |
| **Age** | | | |
| 15–24 | 1 | | |
| 25–34 | 1.31* | 1.04 | 1.66 |
| 35–44 | 1.39* | 1.04 | 1.85 |
| 45+ | 1.02 | 0.69 | 1.49 |
| **Residence** | | | |
| Urban | 1 | | |
| Rural | 0.83 | 0.69 | 1.49 |
| **Wealth Quintile** | | | |
| Poorest | 1 | | |
| Poorer | 1.07 | 0.78 | 1.47 |
| Middle | 1.48* | 1.06 | 2.03 |
| Richer | 1.44* | 1.02 | 2.03 |
| Richest | 1.96*** | 1.30 | 3.00 |
| **Education** | | | |
| No education | 1 | | |
| Primary | 1.32 | 0.93 | 1.89 |
| Secondary | 1.69** | 1.17 | 2.44 |
| Higher | 1.73 | 0.98 | 3.04 |
| **Religion** | | | |
| Christianity | 1 | | |
| Islam | 0.71* | 0.53 | 0.95 |
| Other | 0.10 | 0.53 | 1.89 |
| No religion | 1.05 | 0.68 | 1.61 |
| **Frequency of reading newspaper/magazine** | | | |
| Not at all | 1 | | |
| Less than once a week | 0.84 | 0.63 | 1.11 |
| At least once a week | 0.76 | 0.56 | 1.03 |
| **Frequency of listening to radio** | | | |
| Not at all | 1 | | |
| Less than once a week | 1.02 | 0.75 | 1.38 |
| At least once a week | 1.25 | 0.96 | 1.63 |
| **Frequency of watching television** | | | |
| Not at all | 1 | | |
| Less than once a week | 1.17 | 0.87 | 1.57 |
| At least once a week | 1.23 | 0.93 | 1.62 |
| **Employment** | | | |
| Not working | 1 | | |
| Managerial | 1.06 | 0.64 | 1.76 |
| Clerical | 0.87 | 0.35 | 2.14 |
| Sales | 0.90 | 0.58 | 1.40 |
| Agriculture | 0.84 | 0.57 | 1.24 |
| Services | 0.88 | 0.56 | 1.38 |
| Manual | 0.97 | 0.67 | 1.40 |
| **Survey country** | | | |
| Angola | 1.89* | 0.07 | 3.35 |

(*Continued*)

**Table 2.** (Continued)

| Variables | AOR | 95% CI | |
|---|---|---|---|
| | | **Lower Bound** | **Higher Bound** |
| Burkina Faso | 67.59*** | 8.72 | 523.9 |
| Benin | 1 | | |
| Burundi | 9.87*** | 2.70 | 36.08 |
| DR Congo | 3.03*** | 1.81 | 5.08 |
| Congo | 5.55*** | 3.27 | 9.44 |
| Cote D' Ivoire | 32.10**** | 9.54 | 108.0 |
| Cameroon | 12.93*** | 6.28 | 26.63 |
| Ethiopia | 5.15*** | 0.32 | 1.39 |
| Gabon | 21.40*** | 4.68 | 97.86 |
| Ghana | 4.68** | 1.47 | 14.91 |
| Gambia | 6.77*** | 1.82 | 25.17 |
| Guinea | 6.42*** | 2.95 | 14.00 |
| Comoros | 4.02*** | 1.79 | 9.02 |
| Liberia | 18.71*** | 7.70 | 45.44 |
| Lesotho | 10.46*** | 3.39 | 32.29 |
| Mali | 3.20** | 1.58 | 6.47 |
| Malawi | 7.74*** | 2.01 | 6.31 |
| Mozambique | 3.56*** | 2.01 | 6.31 |
| Nigeria | 5.27*** | 3.00 | 9.26 |
| Namibia | 17.79*** | 2.41 | 131.2 |
| Rwanda | 1.83 | 0.82 | 4.08 |
| Sierra Leone | 4.07*** | 2.26 | 7.34 |
| Senegal | 2.10 | 0.86 | 5.17 |
| Chad | 3.13** | 1.40 | 7.01 |
| Togo | 8.11** | 1.89 | 34.69 |
| Uganda | 6.10*** | 3.07 | 12.11 |
| Zambia | 3.18*** | 1.83 | 5.50 |
| Zimbabwe | 14.37*** | 7.30 | 28.30 |
| Number of Observations | 3353 | | |
| Pseudo R$^2$ | 0.111 | | |

AOR = adjusted Odds Ratio, 95% confidence intervals in brackets

* $p < 0.05$

** $p < 0.01$

*** $p < 0.001$; 1 = reference category.

income, they are less likely to decline a male client who may be reluctant to use a condom [39]. The low rate of consistent condom use for paid sex recorded for some countries in this study could also be attributed to gender-based power distances. Paid sex workers are much constrained in their capacity for condom negotiation with their male clients out of fear of humiliation, violent reprisal, and/ or physical violence [40,41]. Consequently, targeted interventions on condom use messages for men and their paid clients should be context-specific [42]. Self-efficacy training associated with paid sex workers in low-recorded countries with condom usage should be promoted.

The study also revealed that demographic characteristics such as age, wealth quintile, education and religion were independently related to consistent condom use for paid sex among

men in the selected SSA countries. Specifically, consistent condom use among men who pay for sex increases with age, with men aged 35–44 more likely to use condom consistently during paid sex than those aged 15–24. This finding contradicts assumptions from previous studies [31,43,44] that condom use decreases with increasing age among sex workers and their clients. Culturally, men aged between 35–44 years are expected to marry in many SSA countries [44]. Men in these age brackets who fail to do so are perceived as irresponsible in their societies. For younger men in studied countries, the ever-dominant masculinity role orientation in many SSA societies might force them not to consistently wear condoms even for paid sex, despite the associated risk because of their preference for "flesh to flesh" sex [45].

Similar to previous research [31,46–50], wealth quintile and level of education showed the highest odds of consistent condom use among men in the richest wealth quintile and those with secondary level of education paying for sex, compared to their counterparts in the middle wealth quintile as well as those with less and/ or no level of education. Wealth is a common marker of economic status, which is also partly explained by one's educational status. Therefore, men's wealth and secondary educational status may provide them with more exposure to condom prevention campaigns through diverse avenues (e.g., news bulletins, magazines, newspapers, television shows). This exposure may help men to develop condom use negotiating skills during paid sexual encounters. Some previous studies [51–53] have discovered that educated people with affluence are more exposed to diverse media opportunities and educational materials. Such people discuss issues related to HIV/STIs quite frequently with better understanding of transmission modes, have fewer misconceptions, show more positive attitudes toward condom use, and have unbiased assessment of their personal sexual risk. Such persons may be associated with consistent and/or proficient condom use for paid sex [54–56].

Other results further show that Muslim men were less likely to use condom consistently during paid sex, compared to men who were Christians. Possibly, Muslim men from studied countries may oppose any form of birth control measure, including condom use, despite risks associated with casual sex, because of their religious beliefs and other Islamic codes [57–59]. Some studies have already reported lower HIV prevalence within Muslim populations due to the potential biological effect of circumcision that could provide protection against HIV infection [31,59,60]. For circumcised men, the penile shaft is purported to be concealed with a densely keratinized epithelium, offering protection against HIV infection [61]. Till date, the role of genital hygiene regarding male circumcision against HIV infection still remains inconclusive [61]. Future studies should consider consistent condom use among Muslims across the general population to ascertain whether findings may mirror the results obtained with paid sexual partners in SSA.

Additionally, for surveyed countries, men who lived in Burkina Faso had the highest odds of consistent condom use during paid sex, compared to their counterparts in Benin. Burkina Faso is noted for its trans-Africa highway networks; therefore, settlements within those areas may prove to be ideal locations for risky sexual behaviour like paid sex from mobile male migrants. Due to the increased perceived risk of HIV and other STIs, and high vulnerability of female residents, perhaps due to socio-economic reasons, men transiting these settlements are more likely to use condoms consistently for paid sex [39,62]. Cognitive-behavioural interventions should focus more on methods (e.g., social and entrepreneurial training) to minimize vulnerabilities in such areas. Similarly, women of sex business in Benin, especially Cotonou, the capital, are mostly migrants who are less organized, vulnerable, and lack uniform code for their male clients. Hence, the possibility of consistent condom use by men for paid sex may be weakened by competition for more money and survival of their female clients [63]. Strategies related to the impact of condom education may decrease unless women in paid sex work are able to "present a united front" in denying male customers who refuse to use condom [64].

## Practical implications

These findings have implications for relevant interventions. Across many SSA societies, men control sex and condom use. Although women may admit condom use as protective mechanism against diseases, their power to ensure men's continuous use of condom during sexual encounters is questionable because of socio-cultural barriers (e.g., power distances, gendered norms, low social status). Therefore, the implementation of evidence-based interventions (e.g., entrepreneurial skill training, risk reduction counselling-condom negotiation skills) that maintain respect for behavioural and social vulnerabilities, especially among women, are required. These interventions are needed to address the economic realities of women in the sex trade within SSA, especially among women living in areas around the trans-Africa highways. Authorities in SSA should also introduce intervention programmes (e.g., microfinance schemes) to help empower female sex workers by creating alternative sources of livelihood. Men's consistent condom use for paid sex is connected to educational attainment and partly linked with high socio-economic status (i.e., wealth status). Hence, critical efforts are required to encourage men with low educational accomplishment to be more involved with educational campaigns and other capacity building programmes.

However, sex workers may migrate to other regions, especially to places of high economic activity, and their male counterparts may also move to other regions because of job-related tasks. This mobility pattern may limit designed interventions for longer periods. Proposed interventions should be continued because of the likelihood of new entrants who could be exposed to the interventions and would replace targets who have relocated. More innovative methods (e.g., using mobile phones to maintain contact information of men and sex workers and/ or initiate referral links with health professionals) in their new cities would be worthwhile, though daunting. Future studies should use longitudinal design to investigate condom negotiation strategies and the psychological influences these might have on men and female clients as well as test what specific interventions might be more effective.

## Strengths and limitations

Some inherent limitations within the current study have been recognized. The secondary data used for the study exclusively relied on self-reports so might be subject to biases related to social desirability and recall. Because consistent condom use was self-reported, study participants might have over or under reported condom use due to potential stigmatization, especially among populations where sociocultural norms interfere with condom use [64]. Also, recalling consistent condom use for the past year at the time of the survey may reduce the accuracy and consistency of the reports [65]. The cross-sectional nature of the study prohibits causal inferences. Furthermore, the sample size (3,353) limits the generalizability of the findings to the all men who pay for sex within the sub-region since the focus was only on those who had paid for sex in the past 12 months. Despite the noted limitations, using national and regional surveys like DHSs that employed rigorous sampling and data collection procedures provides the methodological justification for current results. Findings provide interpersonal characteristics associated with consistent condom use among men who pay for sex in SSA.

## Conclusions

The study revealed that interpersonal characteristics such as age, wealth quintile, education, religion, as well as surveyed countries were independently related to consistent condom use for paid sex among men. Empirical evidence from this study suggests that consistent condom use during paid sex encompasses complex social and interpersonal characteristics. Other cultural circumstances do influence the decision to use or avoid condoms during sexual

encounters because of varied reasons. For instance, gender and sexual norms in many SSA states promote power distances with exclusive male dominance in matters of sexuality. Hence, many women are coaxed to offer complete subservience to men's sexual need, whether desirable or undesirable. With SSA having the highest sexual and reproductive health burden in the world, continuous application of evidence-based interventions (e.g., educational and entrepreneurial training, micro finance schemes) that account for behavioural and social vulnerabilities are required. Consistent public condom education with messages on socio-cultural barriers to consistent condom use should also be addressed, especially in SSA populations noted with high risk of HIV and other STIs.

## Acknowledgments

We acknowledge Measure DHS for providing us with the data upon which the findings of this study were based.

## Author Contributions

**Conceptualization:** Bright Opoku Ahinkorah, Eugene Budu.

**Data curation:** Bright Opoku Ahinkorah, Eugene Budu, Abdul-Aziz Seidu.

**Formal analysis:** Bright Opoku Ahinkorah, Eugene Budu.

**Methodology:** Abdul-Aziz Seidu, John Elvis Hagan, Jr., Ebenezer Agbaglo, Thomas Hormenu.

**Project administration:** Abdul-Aziz Seidu, Ebenezer Agbaglo, Thomas Hormenu, Thomas Schack, Sanni Yaya.

**Resources:** John Elvis Hagan, Jr., Thomas Schack.

**Visualization:** Abdul-Aziz Seidu, Sanni Yaya.

**Writing – original draft:** John Elvis Hagan, Jr., Thomas Hormenu.

**Writing – review & editing:** Bright Opoku Ahinkorah, Eugene Budu, Abdul-Aziz Seidu, John Elvis Hagan, Jr., Ebenezer Agbaglo, Thomas Hormenu, Thomas Schack, Sanni Yaya.

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
