## [Decision Letter · Decision Letter 0]

12 Jun 2020

PONE-D-20-12659

Predictors of consistent condom use among men who pay for sex in sub-Saharan Africa: Empirical evidence from Demographic and Health Surveys

PLOS ONE

Dear Dr. Hagan Jnr,

Thank you for submitting your manuscript to PLOS ONE. After careful consideration, we feel that it has merit but does not fully meet PLOS ONE’s publication criteria as it currently stands. Therefore, we invite you to submit a revised version of the manuscript that addresses the points raised during the review process.

We look forward to receiving your revised manuscript.

Kind regards,

Joel Msafiri Francis, MD, MS, PhD

Academic Editor

PLOS ONE

Journal Requirements:

2. For studies involving humans categorized by race/ethnicity, age, disease/disabilities, religion, sex/gender, sexual orientation, or other socially constructed groupings, authors should: 1) Explicitly describe their methods of categorizing human populations, 2) Define categories in as much detail as the study protocol allows, 3) Justify their choices of definitions and categories, 4) Explain whether (and if so, how) they controlled for confounding variables such as socioeconomic status, nutrition, environmental exposures, or similar factors in their analysis

The Neurocognition and Action-Biomechanics Research Group, Bielefeld University, Germany

are acknowledged for their financial assistance for the publication of this research.

5. Please include your tables as part of your main manuscript and remove the individual files. Please note that supplementary tables should remain as separate "supporting information" files.

Reviewers' comments:

Reviewer's Responses to Questions

**Comments to the Author**

1. Is the manuscript technically sound, and do the data support the conclusions?

Reviewer #1: Yes

Reviewer #2: Yes

2. Has the statistical analysis been performed appropriately and rigorously? 

Reviewer #1: Yes

Reviewer #2: Yes

3. Have the authors made all data underlying the findings in their manuscript fully available?

Reviewer #1: No

Reviewer #2: Yes

4. Is the manuscript presented in an intelligible fashion and written in standard English?

Reviewer #1: Yes

Reviewer #2: Yes

5. Review Comments to the Author

Reviewer #1: This is a valuable manuscript on predictors of condom use among men who pay for sex in SSA, an important risk group for spread of HIV/STIs. The manuscript should be accepted after some minor revisions:

1. The literature review misses some recent studies on condom use in SSA, from 2018 and 2019.

2. Were there any research questions or hypotheses? None are stated.

3. There are a number of non-native seeming English phrases and also typos throughout. A thorough copy edit is required.

4. There is no discussion of future research implications. Given these findings, what studies need to be done next? What interventions should be tested?

Reviewer #2: Very good article. We welcome such study to understand how to improve condom use among men clients of sex workers.

suggestion: Please update the UNAIDS statistics with the most recent data available at unaided.org.

6. PLOS authors have the option to publish the peer review history of their article (what does this mean?). If published, this will include your full peer review and any attached files.

Reviewer #1: Yes: William D Evans

Reviewer #2: Yes: Bidia Deperthes

---

## [Author Response · Author response to Decision Letter 0]

16 Jun 2020

REBUTTAL LETTER 

This is to acknowledge that the current manuscript has been revised in accordance with the comments from the managing editor and reviewer 2. 

Specific reactions to comments:

1. The manuscript has been revised in accordance with PLOS One requirements

2. Secondary data was used for study analysis. The description of all used variables has been given at the methodology section of the manuscript. 

3. Data for the study is freely accessible at: https://dhsprogram.com/data/available-datasets.cfm. So there are no data restrictions.

4. The funding statement at “Acknowledgments Section” of the manuscript has been removed.

5. The funding information would be provided during the online resubmission.

6. Tables and figure have been inserted in the revised manuscript.

7. Two recent references have been included in the manuscript.

8. We did not state any question or hypothesis but provide the study aim or purpose. This approach is also conventional and acceptable.

9. The manuscript has been proof read to address all the concerns raised by reviewer 1.

10. Practical implications of the study and future research directions have been discussed in the revised manuscript.

11. UNAIDS statistics have been revised in accordance with the comments raised by reviewer 2.

We sincerely appreciate time and effort spent on making this manuscript better for public consumption. We look forward to a positive response.

Thank you.

Yours faithfully,

Dr. John Elvis Hagan Jr.

---

## [Decision Letter · Decision Letter 1]

10 Jul 2020

Predictors of consistent condom use among men who pay for sex in sub-Saharan Africa: Empirical evidence from Demographic and Health Surveys

PONE-D-20-12659R1

Dear Dr. Hagan Jnr,

We’re pleased to inform you that your manuscript has been judged scientifically suitable for publication and will be formally accepted for publication once it meets all outstanding technical requirements.

Kind regards,

Joel Msafiri Francis, MD, MS, PhD

Academic Editor

PLOS ONE

Additional Editor Comments (optional):

Reviewers' comments:

Reviewer's Responses to Questions

**Comments to the Author**

1. If the authors have adequately addressed your comments raised in a previous round of review and you feel that this manuscript is now acceptable for publication, you may indicate that here to bypass the “Comments to the Author” section, enter your conflict of interest statement in the “Confidential to Editor” section, and submit your "Accept" recommendation.

Reviewer #1: All comments have been addressed

Reviewer #2: All comments have been addressed

2. Is the manuscript technically sound, and do the data support the conclusions?

Reviewer #1: Yes

Reviewer #2: Yes

3. Has the statistical analysis been performed appropriately and rigorously? 

Reviewer #1: Yes

Reviewer #2: Yes

4. Have the authors made all data underlying the findings in their manuscript fully available?

Reviewer #1: No

Reviewer #2: Yes

5. Is the manuscript presented in an intelligible fashion and written in standard English?

Reviewer #1: Yes

Reviewer #2: Yes

6. Review Comments to the Author

Reviewer #1: The manuscript is much improved and the authors were responsive to the previous review. The paper should be published after editorial review.

Reviewer #2: Well done. As a condom adviser, this will really be useful to know where to focus on condom programming.

7. PLOS authors have the option to publish the peer review history of their article (what does this mean?). If published, this will include your full peer review and any attached files.

Reviewer #1: No

Reviewer #2: **Yes: **Bidia Deperthes

---

## [Editor Report · Acceptance letter]

17 Jul 2020

PONE-D-20-12659R1 

Consistent condom use among men who pay for sex in sub-Saharan Africa: Empirical evidence from Demographic and Health Surveys 

Dear Dr. Hagan Jr:

I'm pleased to inform you that your manuscript has been deemed suitable for publication in PLOS ONE. Congratulations! Your manuscript is now with our production department. 

Kind regards, 

on behalf of

Dr. Joel Msafiri Francis 

Academic Editor

PLOS ONE